# Association between Antimicrobial Peptide Histatin 5 Levels and Prevalence of *Candida* in Saliva of Patients with Down Syndrome

**DOI:** 10.3390/antibiotics10050494

**Published:** 2021-04-26

**Authors:** Tomoko Komatsu, Kiyoko Watanabe, Nobushiro Hamada, Eva Helmerhorst, Frank Oppenheim, Masaichi Chang-il Lee

**Affiliations:** 1Department of Dentistry for the Special Patient, Kanagawa Dental University, 82 Inaoka-cho, Yokosuka, Kanagawa 238-8580, Japan; komatsu@kdu.ac.jp; 2Yokosuka-Shonan Disaster Health, Emergency Research Center, Kanagawa Dental University, 82 Inaoka-cho, Yokosuka, Kanagawa 238-8580, Japan; 3Department of Oral Microbiology, Kanagawa Dental University, 82 Inaoka-cho, Yokosuka, Kanagawa 238-8580, Japan; watanabe@kdu.ac.jp (K.W.); hamada@kdu.ac.jp (N.H.); 4Department of Molecular and Cell Biology, Boston University Henry M. Goldman School of Dental Medicine, Boston, MA 02118, USA; helmer@bu.edu (E.H.); fropp@bu.edu (F.O.); 5Department of Disaster Related Oral Health, Oxidative Stress/ESR Laboratories, Kanagawa Dental University, 82 Inaoka-cho, Yokosuka, Kanagawa 238-8580, Japan

**Keywords:** small peptides, AMPs, Down syndrome, saliva, histatin 5, *Candida*

## Abstract

There are no studies on *Candida* colonization and micropeptides of saliva in any patient. Therefore, we studied the effects of the salivary antimicrobial peptide histatin 5 on oral fungal colonization; subjects were subdivided into Down syndrome (D) and normal (N) groups by age: N-1 and D-1, age <20 years; N-2 and D-2, age >40 years. Histatin 5 concentration in saliva was measured by enzyme-linked immunosorbent assay. Oral *Candida* species were identified using CHROMagar *Candida*. *Candida* colonization was significantly enhanced in the D-1 and D-2 groups compared to the N-1 and N-2 groups. There was no predominant difference in salivary histatin 5 concentration between the D-1 and N-1 groups, but it was significantly lower in the D-2 group than in the N-2 group. Only in the N-2 group was there a correlation between the concentration of histatin 5 and total protein, while no correlation was found in the other groups. In elderly patients with Down syndrome, the decrease in histatin 5 shown in this study may lead to oral *Candida* colony formation. Therefore, the results of this study suggest that a deficiency of the antimicrobial peptide histatin 5 could possibly induce oral *Candida* infection in DS.

## 1. Introduction

Down syndrome (DS), an autosomal abnormality, is a disorder caused by trisomy of chromosome 21 [1] and displays many functional and physical characteristics including intellectual impairment, congenital heart disease, leukemia, dementia, and increased susceptibility to infection (especially respiratory infections and periodontal disease) [2,3]. DS is also characterized by immune system malfunctions including cell responses [4], humoral immunity, and inflammatory cells.

Much of the normal microbiota present in the oral cavity becomes pathogenic when the host’s defense system, i.e., the immune system, is weakened. The typical oral candidiasis that occurs in such cases is an opportunistic infection [5]. The prevalence of *Candida* in the oral cavity has been reported to be 45–65% in normal children [6] and 30–45% in normal adults [5]. In particular, *C. albicans* is a potent endogenous fungus, and 26–44% of healthy individuals without mucosal disease have been found to be oral carriers of *C. albicans* [7].

Saliva plays an important role in preventing the progression from microbial colonization to infection in the oral cavity by the innate immune response and nonimmune host defense system [8]. Therefore, saliva not only contains numerous antimicrobial proteins and peptides, but also has buffering capacity, which is important in the oral defense against infections. Histatin 5 is a major component of a small protein family produced by the parotid and submandibular glands [9]. Histatin 5 exhibits potent antifungal activity, including against *C. albicans* [9] and other pathogenesis-associated *Candida* species [10].

Patients with DS are known to be susceptible to fungal infections, and clinical studies have been conducted related to the *Candida* fungal population in the oral cavity [11]; however, no in vivo studies have been conducted to assess the levels of histatin 5 in patients with DS. Therefore, we investigated the association between salivary levels of oral histatin 5 and the *Candida* possession rate by comparing children and adults with DS and age- and sex-matched controls without DS. Therefore, we investigated the association between salivary levels of oral histatin 5 and *Candida* carriage rates by comparing children and adults with DS and normal controls without DS to understand the physiological role of histatin 5 in age- and gender-matched healthy individuals.

## 2. Results

### 2.1. Prevalence of Candida Species Colonization on the DS and N Groups

The percentage of *Candida* carriers among the participants is shown in Figure 1. The prevalence of colonization and the total number of *Candida* species in the central area of the dorsal surface of the tongue were observed as follows: *C. albicans* was the most common isolate, with seven (29.2%) in group N-1, one (4.2%) in group N-2, 13 (65.0%) in group DS-1, and 23 (74.2%) in group DS-2 (Figure 1). On the other hand, there were only one (4.2%) *C. tropicalis* carriers in group N-1, two (10.0%) in group DS-1, and six (19.4%) in group DS-2 (Figure 1). There was one carrier (4.2%) of *C. krusei* in group N-2, one (5.0%) in group DS-1, and six (19.4%) in group DS-2 (Figure 1). *C. krusei* was not recovered from the N-1 group (Figure 1). *C. tropicalis* was not recovered from the N-2 group (Figure 1). Fungal cultures showed differences in both the colonization of *Candida* species and the prevalence of *Candida* carriers between the N-2 and DS-2 groups.

### 2.2. Comparison of Salivary Flow Rates between the DS and N Groups

Figure 2 shows a comparison of salivary flow rate between DS and N groups. Statistical analysis revealed that the N-1 group had a significantly higher mean unstimulated salivary flow rate than the N-2 group, and the N-2 group had a significantly higher mean unstimulated salivary flow rate than the DS-2 group (*p* < 0.01; Figure 2); in addition, there was a significant difference between the DS-1 group and the DS-2 group (*p* < 0.01; Figure 2). There was no difference between the N-1 and DS-1 groups (Figure 2).

### 2.3. Comparison of Total Salivary Protein Concentrations between the DS and N Groups

Statistical analysis showed that the N-1 group had a significantly lower mean salivary protein concentration than the N-2 group (*p* < 0.01). Statistical analysis also showed that the N-1 group had a significantly lower mean salivary protein concentration than the DS-1 group (*p* < 0.01; Figure 3). Furthermore, statistical analysis showed that the mean total salivary protein concentration was significantly lower in the N-2 group compared to the DS-2 group (*p* < 0.01; Figure 3). On the other hand, there was no difference between the DS-1 and DS-2 groups (Figure 3).

### 2.4. Comparison of Salivary Histatin 5 Concentrations between the DS and N Groups

Statistical analysis showed that the mean salivary concentration of histatin 5 was significantly lower in the N-1 group than in the N-2 group (*p* < 0.01; Figure 4), and the mean salivary concentration of histatin 5 was significantly lower in the DS-1 group than in the DS-2 group (*p* < 0.05; Figure 4). The mean salivary concentration of histatin 5 was significantly higher in the N-2 group compared to the DS-2 group (*p* < 0.01; Figure 4). However, there was no difference between the N-1 and DS-1 groups (Figure 4).

### 2.5. Correlation with Histatin 5 and Total Salivary Protein Concentration

The salivary concentration of histatin 5 was strongly correlated with the total salivary protein concentration only in the N-2 group (Figure 5 and Figure 6, Table 1), and no correlation was observed among the other groups (Figure 6, Table 1).

## 3. Discussion

The oral cavity is a rich habitat for microorganisms such as bacteria, fungi, and viruses, which are in good harmony with their human hosts. The relationship between microorganisms and the human host seems to be related to the immune status of each individual. The *Candida* carrier rate was higher in healthy children and adolescents (N-1 group) compared to healthy adults (N-2 group) (Figure 1). Children’s immunity matures as they grow; thus, there is a possibility that *Candida* levels will increase when their immunity is still low [12]. Contrary to the results of group N, the proportion of *Candida* carriers was high in all cases of DS children, adolescents, and adults. Here, 65.0% of the children and students with DS were infected with one or more species of *Candida*, with *C. albicans* being the most commonly identified, followed by *C. tropicalis* and *C. krusei* (Figure 1). The results for these children and adolescents confirmed similar results to previous reports. According to [13], in DS patients aged 3–22 years, 74% of the samples taken from the dorsal side of the tongue had *Candida* species and 84% of the culture-positive samples had *C. albicans* strains as the most common, followed by *C. tropicalis*. In another study, 69% of children and adolescents with DS were reported to have *C. albicans* identified in samples taken from their cavity [11]. Another compelling result is that the highest numbers of fungi are observed in patients with geographic and fissured tongues [14], which is found in 38% of DS patients [13]. These reports suggest that susceptibility to oral infections such as candidiasis is increased in DS from an early childhood.

Compared to the N group, the DS group had lower unstimulated salivary flow rate from early childhood, which decreased with age (Figure 2). In this study, it was also confirmed that salivary flow rate was decreased in DS patients (Figure 2). The decrease in salivary flow rate in DS may be due to progressive salivary gland dysfunction related to saliva secretion in DS compared to normal subjects. These results suggest that the mechanism of salivary gland dysfunction in DS from an early childhood requires further study.

Salivary protein concentrations were higher among DS subjects than among healthy normal subjects of the same age; there was no difference between the DS-1 and DS-2 groups (Figure 3). From these results, there is the possibility that saliva protein synthesis is enhanced from early childhood in DS. In terms of protein concentration in the saliva of DS subjects, there was no difference between the DS-1 and DS-2 groups, but it was higher than that of normal controls of the same age (Figure 3). These results suggest that salivary protein synthesis may be enhanced in DS from an early childhood. It is known that most of the sources of proteolytic enzymes in total saliva are derived from oral microbiota associated with periodontopathic bacteria [15]. It has been reported that not only children with DS but also adults may have high numbers of periodontal bacteria [16]. Since proteolysis in oral fluid serves to liberate small peptides from larger, less active, or inactive precursor proteins, the high protein concentrations in DS may be due to the denaturation of proteins in saliva by proteolysis [17].

In the ELISA study, the concentration of histatin 5 was comparable between DS and healthy subjects (Figure 4). In agreement with our study, salivary proteomics has shown that salivary levels of the antimicrobial peptide histatin 5 were similar in patients with DS aged 10–17 years and in healthy normal groups [18]. However, the salivary flow rate of DS was two times slower than that of N-1(Figure 2), which may be due to the action of consistently replenishing intact histatin 5 in the oral cavity to counteract the proteolysis of histatin 5 in DS. It has been reported that children with DS may have different compositions of biochemical components in saliva due to changes in the metabolism of salivary gland ducts and gland cells [19]. This modulation of histatin 5 and other biochemical components may also influence the prevalence of *Candida* in DS children.

In the adult group, the concentration of histatin 5 was significantly lower in the DS-2 group than in the N-2 group (*p* < 0.01, Figure 4), which was different from the results obtained in the younger group. In addition, the results of fungal culture showed that colony formation of *Candida* was significantly higher in the DS-2 group than in the N-2 group (Figure 1). In DS-2 compared to N-2, the amount of histatin 5 in total saliva decreased with reducing salivary flow rate (Figure 2). Due to the bacterial count and inflammatory burden in DS patients, the prevalence of periodontitis was characterized by a marked increase with age. [2]. Since the oral microbial imbalance in the oral cavity of the DS-2 group with low salivary flow may be a major local factor in the increase of *Candida* carriers in DS patients (Figure 1), total salivary proteolysis by oral bacteria-derived proteases may rapidly degrade histatin 5 (Figure 5 and Figure 6).

These results are consistent with a previous study of immunocompromised patients by Khan et al. [20], which showed lower histatin 5 levels in HIV-positive individuals and a higher prevalence of *C. albicans* in these patients. In such immunocompromised individuals, the innate immune function involved in histatin 5 secretion may be reduced due to decreased saliva volume or increased oral microbiota, or it may not be activated due to insufficient protective response. Therefore, in immunocompromised adults and patients with DS, decreased histatin 5 levels in saliva may be related to increased oral *Candida* concentrations. Furthermore, the results of fungal culture showed a slight increase in the rate of *Candida* carrier in the DS-2 group compared to the DS-1 group (Figure 1).

Histatin 5 levels were elevated in the DS-2 group (Figure 4). However, the difference in histatin 5 concentration between DS-1 and DS-2 was smaller than that between N-1 and N-2. In addition to a deficiency of immunoglobulins in saliva, an inadequate increase in histatin 5 may be partially responsible for the increased rate of *Candida* carriers in adults with DS [21]. It has already been shown that one or more salivary glands in the area of bilateral parotid and submandibular glands are absent in some DS children [19]. Decreased salivary function of the parotid gland was also reported in subjects with DS [22]. Since histatin 5 is mainly secreted by the parotid gland, the decrease in histatin 5 concentration in saliva (Figure 4) can be explained by the greater decrease in salivary secretion. These results suggest that histatin 5 is expressed in the saliva of DS, indicating that DS has dysfunctional salivary glands. In the oral cavity, it is well known that the adsorption of proteins to oral structures contributes to the decreased concentration of histatin 5 in expectorated samples [23]. It was reported that calcium levels were significantly higher in children with DS and dental caries compared to controls with and without dental caries [24]. Calcium ions are thought to be involved in binding to histatin 5, lowering the concentration of histatin 5 and inhibiting the antifungal effect of histatin 5 [25].

Furthermore, there was a correlation between histatin 5 concentration and total protein only in the N-2 group (Figure 5 and Figure 6, Table 2), while no association was found between histatin 5 concentration and total protein in the other groups (Figure 6, Table 2). Innate oral defense by salivary histatin 5 in healthy individuals plays a role in susceptibility to oral candidiasis. These results may be associated with increased *Candida* colony formation associated with histatin 5 in saliva, at least in healthy adults. Therefore, our results suggest that adaptive immunity is impaired in the oral cavity of DS patients during childhood and adulthood. Subtle abnormalities in both humoral and cellular immune responses involved in histatin 5 levels are related to genes encoded on chromosome 21 and may affect a large number of carriers of DS. Healthy infants have an adaptive immune system that matures with age, but their defenses against infections tend to be weak. However, in DS, an effective immune response is not fully developed, even in the elderly. The mechanism is different between people with DS and normal people; people with DS may have lower immune defenses from childhood and may not be able to raise their immune defenses as adults. We need to study in the future whether the physicochemical conditions of saliva, tolerance to histatin 5, and dietary differences may be related to the results of this study.

## 4. Materials and Methods

### 4.1. Study Population

The participants in this study had DS with karyotype 47 XX, 21+ (females) and 47 XY, 21+ (males) and were in good general health. Exclusion criteria were treatment with antibiotics and steroids within the past 2 months, use of nonsteroidal anti-inflammatory drugs and antifungals within the past 3 months, use of corticosteroids for asthma, use of oral topical corticosteroids within the past 3 months, partial or complete dental prosthesis, history of chemotherapy or radiotherapy, and history of diabetes, hepatitis B and C infection, congenital immunodeficiency syndrome, diabetes, human immunodeficiency virus (HIV) infection, and acquired immunodeficiency syndrome. This study was approved by the Institutional Ethics Committee (No. 169) of the Kanagawa Dental University in accordance with the Declaration of Helsinki. In addition, the participants or their legal representatives were informed about the study, and informed consent was obtained from all voluntary participants or their legal representatives.

A total of 51 DS patients (34 males, 17 females, age range 5–62 years) and 48 normal controls (20 males, 28 females, age range 8–57 years) were included in the study.

The DS and N groups were subdivided according to age, with the first group consisting of DS (DS-1) and normal (N-1) subjects under 20 years of age, and the second group consisting of DS (DS-2) and normal (N-2) subjects over 40 years of age. The sex composition and mean age of the study groups are shown in Table 1.

Oral evaluation, sample collection for fungal culture, and unstimulated saliva collection were performed by oral healthcare professionals.

### 4.2. Saliva Collection and Measurement of the Salivary Flow Rate

Unstimulated whole saliva was collected by using the absorbent method [26]. Salivary samples were collected between 10:00 a.m. and 2:00 p.m. and at least 1 h after food ingestion and prior to clinical measurements due to the difficulty in collecting stimulated saliva from individuals with intellectual disabilities. One neutral, uncovered, absorbent cotton roll (Salivette^®^, Sarstedt, Nümbrecht, Germany) was placed into the mouth under the tongue for exactly 5 min. To determine the salivary flow rate, the cotton rolls were weighed before and after saliva collection using an electronic scale sensitive to 0.01 g. The weight gain during the 5 min interval was then converted into mL·min^−1^ of saliva [27]. To determine the salivary flow rate, cotton rolls were weighed before and after saliva collection using an electronic scale sensitive to 0.01 g. The weight gain during the 5 min interval was converted to mL·min^−1^ of saliva [27]. Immediately after weighing, saliva samples were clarified by centrifugation at 10,000× *g* for 15 min at 4 °C [28]. The supernatants were stored at −40 °C for up to 1 month prior to analysis.

### 4.3. Measurement of Salivary Antimicrobial Peptide

The levels of histatin 5 in saliva samples, a salivary antimicrobial peptide [9], were determined using an enzyme-linked immunosorbent assay (ELISA) kit (Wuhan Eiaab Science Co., Ltd.; Wuhan, China). The microtiter plate was precoated with an antibody specific for natural human histatin 5 (Wuhan Eiaab Science Co., Ltd; Wuhan, China). No substantial cross-reactivity or interference was observed. Standards and samples (100 μL each) were added to the microtiter plate wells in duplicate and incubated at 37 °C for 2 h. After incubation, 100 μL of a biotin-conjugated polyclonal antibody solution preparation specific for histatin 5 was added to each well and incubated at 37 °C for 1 h. After washing away any unbound substances, 100 μL of horseradish peroxidase-conjugated avidin solution was added to each well for 1 h at 37 °C. Then, 90 μL of substrate solution was added to each well and reacted for at 37 °C for 30 min. The enzyme-substrate reaction was terminated by the addition of 50 μL of sulfuric acid solution, and the color change was measured spectrophotometrically at a wavelength of 450 nm. Only wells containing histatin 5, biotin-conjugated antibody, and enzyme-conjugated avidin showed a color change. The concentration of histatin 5 in the samples was calculated from the standard curve.

Total salivary protein concentration was determined with the bicinchoninic acid (BCA) method [29] using a Micro BCA Protein Assay Kit (#23235, Thermo Fisher Scientific K.K, Kanagawa, Japan) according to the manufacturer’s instructions.

### 4.4. Isolation and Identification of Candida Species

Oral *Candida* samples were collected from behind the center of the dorsal surface of the tongue by swabbing; the swab was rotated thoroughly in one direction with back and forth movement. The cotton swabs were placed in 1 mL of transport medium composed of 5.0 mg of NaCl, 1.1 mg of Na_2_HPO_4_, 0.09 mg of CaCl_2_, 1.5 mg of thioglycolic acid sodium salt, and 5.6 mg of agar (pH 7.2; 0.4% thioglycolate/0.15% phosphate-buffered saline) in sterile bottles (Seedswab No. 1, Eiken Chemical Co., Ltd., Tokyo, Japan) [30]. These were suspended in 1 mL of physiological saline to facilitate dissolution of the swabs and improve recovery of the oral microbiota. After agitating a cotton swab in 1 mL of physiological saline, 100 μL of the sample was aerobically inoculated onto CHROMagar Candida (Becton Dickinson, Sparks, MD, USA) at 37 °C for 48 h. Three *Candida* species (*C. albicans*, *C. tropicalis*, and *C. krusei*) were identified on CHROMagar on the basis ofcolony morphology and pigmentation by following the manufacturer’s instructions and the description by Tan and Peterson [31]. *C. albicans*, *C. tropicalis,* and *C. krusei* produced green, blue, and pink and fuzzy colonies, respectively. Colonies of non-*C. albicans* species were certified by using the API 20C AUX system for yeasts (bioMérieux VITEK, Tokyo, Japan).

### 4.5. Statistical Analysis

Descriptive data including means and standard deviations were determined for each parameter in each group and were used for analysis. One-way analysis of variance (ANOVA) was used for multigroup comparisons followed by Scheffe’s test for pairwise comparisons. The correlation between concentrations of total salivary protein and salivary histatin 5 concentrations was analyzed by Pearson’s correlation test. For all tests, a *p*-value of 0.05 or less was considered statistically significant. Statistical analysis was performed using SPSS software version 23 (IBM SPSS Statistics, Tokyo, Japan).

## 5. Conclusions

In conclusion, the colony formation of *Candida* was significantly higher in the DS group than in the normal group. Histatin 5 levels did not differ between the DS group and normal children and adolescents, but were significantly lower in adults in the DS group compared to the normal group. The progressive colonization of *Candida* in people with DS contributes to the inherent nature of the vulnerable population. Therefore, it is important to prevent oral infections caused by *Candida* by initiating appropriate oral care in DS patients at an early age.

## Figures and Tables

**Figure 1 antibiotics-10-00494-f001:**
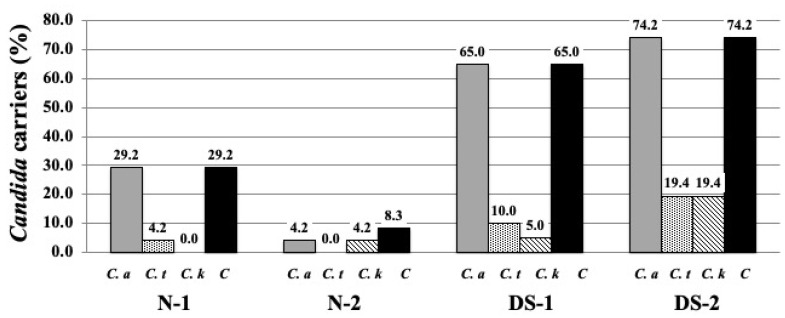
The distributions of *Candida* carriers by percentage in the study group and normal group. *Candida* species were isolated from the center of the dorsal surface of the tongue from individuals with Down syndrome (DS) and healthy normal (N) individuals. The DS and N groups were subdivided according to age: group 1 consisted of DS (DS-1) and normal (N-1) subjects under 20 years of age, and group 2 consisted of DS (DS-2) and normal (N-2) subjects over 40 years of age, as shown in Table 1. The distributions of *Candida* carriers by percentage in the DS group and N group are shown. The gray bars represent the distributions of *C. albicans* carriers by percentage in the study group and normal group. The dotted bars represent the distributions of *C. tropicalis* carriers by percentage in the study group and normal group. The cross-hatched bars represent the distributions of *C. krusei* carriers by percentage in the study group and normal group. The black bars represent the distributions of the total number of *Candida* carriers by percentage in the study group and normal group. *C. a: C. albicans, C. t: C. tropicalis, C. k: C. krusei, C: total number of Candida.*

**Figure 2 antibiotics-10-00494-f002:**
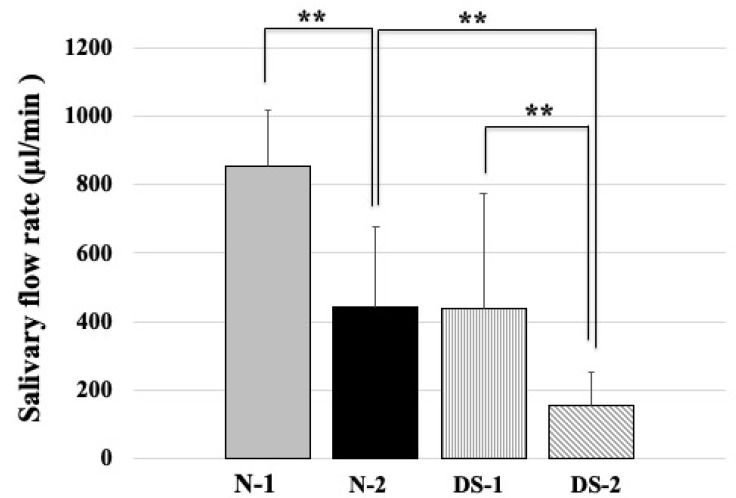
Comparison of total salivary protein concentrations between the Down syndrome (DS) and normal (N) groups. The DS and N groups were subdivided according to age, with the first group consisting of DS (DS-1) and normal (N-1) subjects under 20 years of age, and the second group consisting of DS (DS-2) and normal (N-2) subjects over 40 years of age. The sex composition and mean age of the study groups are shown in Table 1. One-way ANOVA test results of comparing salivary flow rate in four groups and the pairwise comparison using Scheffe’s test for all four groups. ** *p* < 0.01; F-value 38.91.

**Figure 3 antibiotics-10-00494-f003:**
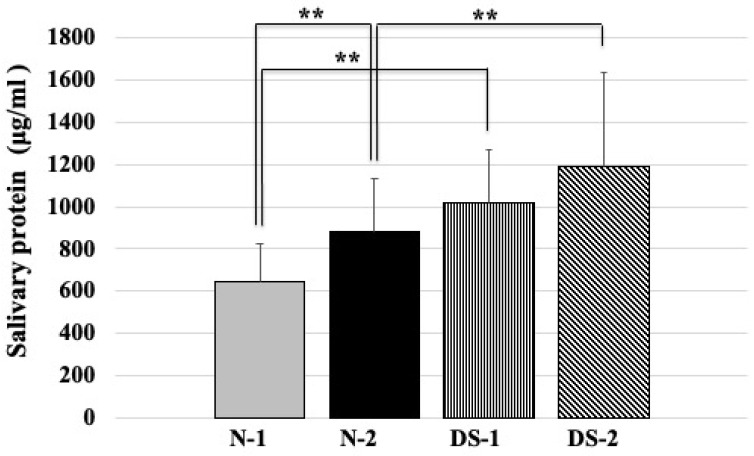
Comparison of total salivary protein concentrations between the Down syndrome (DS) and normal (N) groups. The DS and N groups were subdivided according to age, with the first group consisting of DS (DS-1) and normal (N-1) subjects under 20 years of age, and the second group consisting of DS (DS-2) and normal (N-2) subjects over 40 years of age. The sex composition and mean age of the study groups are shown in Table 1. One-way ANOVA test results of comparing salivary protein concentration in four groups and the pairwise comparison using Scheffe’s test for all four groups. ** *p <* 0.01; F-value 14.11.

**Figure 4 antibiotics-10-00494-f004:**
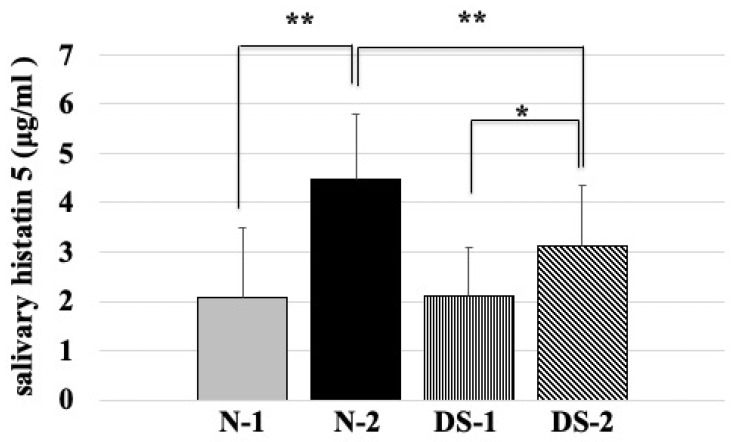
Comparison of salivary histatin 5 concentrations between the Down syndrome (DS) and normal (N) groups. The DS and N groups were subdivided according to age, with the first group consisting of DS (DS-1) and normal (N-1) subjects under 20 years of age, and the second group consisting of DS (DS-2) and normal (N-2) subjects over 40 years of age. The sex composition and mean age of the study groups are shown in Table 1. One-way ANOVA test results of comparing salivary histatin 5 concentration in four groups and the pairwise comparison using Scheffe’s test for all four groups. ** *p <* 0.01, * *p <* 0.05; F-value 16.59.

**Figure 5 antibiotics-10-00494-f005:**
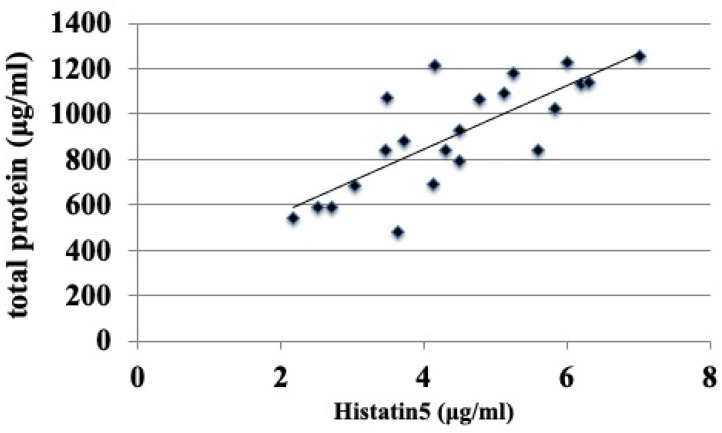
Correlation with histatin 5 and total salivary protein concentration in normal (N-2) subjects. The N-2 groups were normal (N-2) subjects over 40 years of age. Each rhombus is the histatin 5 concentration relative to the salivary protein concentration in N-2 groups. The N (normal) groups were subdivided according to age: normal (N-2) subjects over 40 years of age as shown in Table 1. The line yielded by a linear regression is plotted on the graph.

**Figure 6 antibiotics-10-00494-f006:**
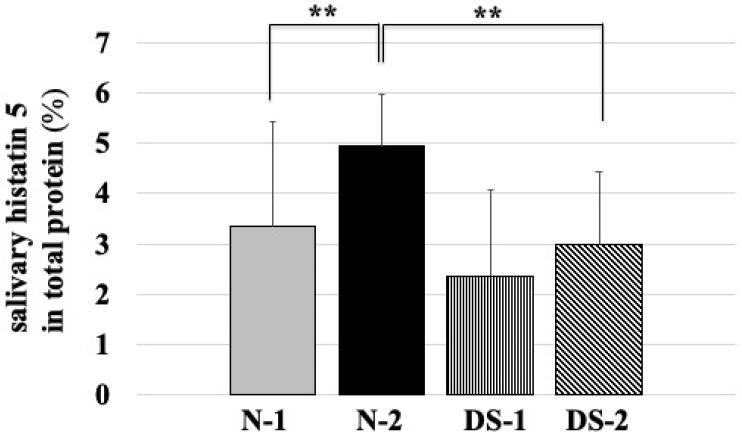
*Comparison of* salivary histatin 5 content in total protein between the Down syndrome (DS) and normal (N) groups. The DS and N groups were subdivided according to age, with the first group consisting of DS (DS-1) and normal (N-1) subjects under 20 years of age, and the second group consisting of DS (DS-2) and normal (N-2) subjects over 40 years of age. The sex composition and mean age of the study groups are shown in Table 1. One-way ANOVA test results of comparing salivary histatin 5 content in total protein in four groups and the pairwise comparison using Scheffe’s test for all four groups. ** *p <* 0.01; F-value 9.31.

**Table 1 antibiotics-10-00494-t001:** The gender composition and mean ages of the study group.

	Under 20 Years Old		Over 40 Years Old	
	N-1	DS-1	N-2	DS-2
**Male**	12	12	7	22
**Female**	12	8	17	9
**Total**	24	20	24	31
**Average of age (SD)**	8.5 ± 2.0	11.3 ± 4.2	47.1 ± 4.9	48.9 ± 6.5

**Table 2 antibiotics-10-00494-t002:** Pearson’s correlation coefficient test comparing the salivary histatin 5 to total salivary protein concentration among the groups.

Study Group	Correlation Coefficient
N-1	0.08134
N-2	0.77589 **
DS-1	0.40228
DS-2	0.14843

** Significant at 0.01 level (*p* < 0.01).

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
