# Peer review of "Association between Antimicrobial Peptide Histatin 5 Levels and Prevalence of *Candida* in Saliva of Patients with Down Syndrome"

_antibiotics, 2021, doi:10.3390/antibiotics10050494_

Round 1
Reviewer 1 Report
The manuscript entitled "A study of antifungal effects of an antimicrobial peptide in the saliva of patients with Down syndrome" by Komatsu et al. could be an excellent resource for the researchers. Despite the interesting topic, authors should undertake minor revision to make it appropriate for publication in Antibiotics Journal.
Abstract need to be re-written so as to highlight the importance of the study. Important observations and discussion needed why predominance of histatin was observed in a certain groups and why not in others.
In the introduction section, the Authors wrote “potent bactericidal activity against pathogenic fungi, including C. albicans [8] and other 57 pathogenesis-associated Candida species [9]”. I fail to understand why the term antibacterial is used for effects observed against fungus.
The term “Candida carriers” means what in the last para of the introduction.
Please discuss What controls were used for the study?
In discussion, the Author mentions “In this study, it was also confirmed that salivary flow 138 rates was decreased in DS patients” I fail to understand when authors are reporting something, why the reference to literature is given (even if no correlation of the results is done).
Have the author's checked the proteolysis of protein in the saliva? If yes, what results were inferred from the study?
Authors need to polish the results and make them concise so as to bring clarity to the thoughts.
I believe this manuscript can turn out to be quite significant by updating it for recent findings and increasing the scope.
Besides, improve the quality of figures.
Author Response
In accordance with the suggestion of this MDPI Antibiotics Editorial Office, we inserted line number, and rewritten or added parts are indicated by yellow highlighting in the revised manuscript.
Reviewer 1:
Abstract need to be re-written so as to highlight the importance of the study. Important observations and discussion needed why predominance of histatin was observed in a certain groups and why not in others.
According to the suggestion of this reviewer, we have rewritten the abstract in the revised manuscript (line 51-54).
In the introduction section, the Authors wrote “potent bactericidal activity against pathogenic fungi, including C. albicans [8] and other 57 pathogenesis-associated Candida species [9]”. I fail to understand why the term antibacterial is used for effects observed against fungus.
According to the suggestion of this reviewer, we have rewritten the introduction in the revised manuscript (line 76).
The term “Candida carriers” means what in the last para of the introduction.
Please discuss What controls were used for the study?
According to the suggestion of this reviewer, we have rewritten the introduction in the revised manuscript (line 85-86).
In discussion, the Author mentions “In this study, it was also confirmed that salivary flow 138 rates was decreased in DS patients” I fail to understand when authors are reporting something, why the reference to literature is given (even if no correlation of the results is done).
According to the suggestion of this reviewer, we have deleted the indicated references in the revised manuscript.
Have the author's checked the proteolysis of protein in the saliva? If yes, what results were inferred from the study?
We did not perform experiment with proteolysis of protein in this study, but the co-authors have already reported (J Dent Res. 2011 Nov; 90(11): 1325–1330. doi: 10.1177/0022034511420721, PMCID: PMC3188460, FASEB J. 2009 Aug; 23(8): 2691–2701. doi: 10.1096/fj.09-131045 PMCID: PMC2717783).
Authors need to polish the results and make them concise so as to bring clarity to the thoughts.
According to the suggestion of this reviewer, we have rewritten all results section in the revised manuscript.
I believe this manuscript can turn out to be quite significant by updating it for recent findings and increasing the scope.
Besides, improve the quality of figures.
We really agree with this suggestion from the reviewer and add the text (line 232-234), and have also made improved the quality of all figure.

Reviewer 2 Report
The manuscript titled “A study of antifungal effects of antimicrobial peptide in saliva of patients with Down syndrome” deals with the description of oral Candida composition in patients with Down syndrome vs heathy individuals. Moreover, authors try to correlate these data with histatin 5 concentrations in their saliva. For that, authors just measure Candida CFUs, salivary flow rate, total proteins in saliva and histatin 5 levels. This is an interesting research that may be useful for medicinal microbiology research. These considerations make the paper interesting for the readership of the journal Antibiotics.
Regarding the scientific methods, I miss some data that possibly authors will not be able to achieve unless you restart this research:
- A genetic (i.e. 16S rRNA based) identification of the isolates
- pH measurements of the oral secretions
- History of oral injuries
Despite this, the manuscript is well organized. However, there are some aspects that may be addressed. Therefore, I suggest some revisions that may improve the manuscript or are necessary to fully understand this research.
Title: This title seems broader than the research. I suggest being more concise. I.e.: Use histatin 5 instead of antimicrobial peptide (or just concise, the antimicrobial peptide histatin 5). Rather than saying “study of antifungal effects (…) in saliva”, I would also concise this to “…association between histatin 5 levels and prevalence of Candida in saliva….”.
Abstract:
- Line 34. “due to”: Based on your data, you can not declare that your Candida colonization is due to histatin deficiency, but just hypothesize. Please, include “may be”, “possible” or some similar expression within the sentence.
Introduction section
- Line 45. The terms “flora” or “microflora” are incorrect and outdated. It was applied when no-body knew about the real microbiota and it refers to some kind of “flora” that does not reflects the real characteristics of the microbiota. Please, change it for “bacterial communities”, “microbiota” or something different. Do the same in line 148.
- Line 46-47. Please, add a reference for the assumption of being an opportunistic infection.
- Line 53-54. Add reference for this sentence.
- Line 59. “using fungal colonies” This expression sound rare and not concise, please, change it to a more precise sentence.
Results section
- Please, be consistent with the use of “N” or “C” for control groups, you have both terms in the manuscript text, in tables and in Figure 1.
Discussion section
- Lines 142-147. These lines are too repetitive. It is said the same in line 143 and in line 145-146. The same with line 144 and line 147. Please, improve the text in this sense.
- Line 168. Remove first “oral”, which is repetitive.
- Line 174. Please, italize the species.
- Lines 201-202. Please, add reference for this sentence
- There are many research articles in which the Candida prevalence, salivary flow rate and similar aspects have been addressed for DS patients. I miss more comparisons of your results with previous results.
- I also miss some discussion for other possible reasons that may explain your results. ¿May your results be due to different physicochemical conditions in saliva? ¿may be resistance to histatin implicated? ¿may be different dietary habits related with your results?
Author Response
In accordance with the suggestion of this MDPI Antibiotics Editorial Office, we inserted line number, and rewritten or added parts are indicated by yellow highlighting in the revised manuscript.
Reviewer 2
The manuscript titled “A study of antifungal effects of antimicrobial peptide in saliva of patients with Down syndrome” deals with the description of oral Candida composition in patients with Down syndrome vs heathy individuals. Moreover, authors try to correlate these data with histatin 5 concentrations in their saliva. For that, authors just measure Candida CFUs, salivary flow rate, total proteins in saliva and histatin 5 levels. This is an interesting research that may be useful for medicinal microbiology research. These considerations make the paper interesting for the readership of the journal Antibiotics.
Regarding the scientific methods, I miss some data that possibly authors will not be able to achieve unless you restart this research:
A genetic (i.e. 16S rRNA based) identification of the isolates
pH measurements of the oral secretions
History of oral injuries
Despite this, the manuscript is well organized. However, there are some aspects that may be addressed. Therefore, I suggest some revisions that may improve the manuscript or are necessary to fully understand this research.
Minor comments:
Title: This title seems broader than the research. I suggest being more concise. I.e.: Use histatin 5 instead of antimicrobial peptide (or just concise, the antimicrobial peptide histatin 5). Rather than saying “study of antifungal effects (…) in saliva”, I would also concise this to “…association between histatin 5 levels and prevalence of Candida in saliva….”.
According to the suggestion of this reviewer, we have rewritten the title in the revised manuscript (line3-4).
Abstract:
Line 34. “due to”: Based on your data, you can not declare that your Candida colonization is due to histatin deficiency, but just hypothesize. Please, include “may be”, “possible” or some similar expression within the sentence.
According to the suggestion of this reviewer, we have rewritten the abstract in the revised manuscript (line 51-54).
Introduction section
Line 45. The terms “flora” or “microflora” are incorrect and outdated. It was applied when no-body knew about the real microbiota and it refers to some kind of “flora” that does not reflects the real characteristics of the microbiota. Please, change it for “bacterial communities”, “microbiota” or something different. Do the same in line 148.
According to the suggestion of this reviewer, we have changed all the words "flora" and "microflora" to "microbiota (line 63, 164, 195, 303).
Line 46-47. Please, add a reference for the assumption of being an opportunistic infection.
According to the suggestion of this reviewer, we have added refrences 5 in the revised manuscript (line 65).
Line 53-54. Add reference for this sentence.
According to the suggestion of this reviewer, we have added refrences 8 in the revised manuscript (line72).
Line 59. “using fungal colonies” This expression sound rare and not concise, please, change it to a more precise sentence.
According to the suggestion of this reviewer, we have rewritten the title in the revised manuscript (line79).
Results section
Please, be consistent with the use of “N” or “C” for control groups, you have both terms in the manuscript text, in tables and in Figure 1.
According to the suggestion of this reviewer, we have changed all the words "C" a to "N" in the manuscript text, in tables and in Figure 1 of the revised manuscript.
Discussion section
Lines 142-147. These lines are too repetitive. It is said the same in line 143 and in line 145-146. The same with line 144 and line 147. Please, improve the text in this sense.
According to the suggestion of this reviewer, we have deleted the repetitive text in the revised manuscript.
Line 168. Remove first “oral”, which is repetitive.
According to the suggestion of this reviewer, we removed “oral” in the revised manuscript.
Line 174. Please, italize the species.
According to the suggestion of this reviewer, we have made italize the species in the revised manuscript (line143).
Lines 201-202. Please, add reference for this sentence
According to the suggestion of this reviewer, we have added refrences 19 in the revised manuscript (line 178).
There are many research articles in which the Candida prevalence, salivary flow rate and similar aspects have been addressed for DS patients. I miss more comparisons of your results with previous results.
I also miss some discussion for other possible reasons that may explain your results. ¿May your results be due to different physicochemical conditions in saliva? ¿may be resistance to histatin implicated? ¿may be different dietary habits related with your results?
We really agree with this suggestion from the reviewer and would like to proceed with further research and have added to the discussion to include research on the mechanisms involved in this study (line232-234).
